# Design and synthesis of broadband absorption covalent organic framework for efficient artificial photocatalytic amine coupling

Yuanding Fang[1,2,3], Youxing Liu [2,3], Haojie Huang[2], Jianzhe Sun[2], Jiaxing Hong[2], Fan Zhang[2], Xiaofang Wei [2], Wenqiang Gao[2], Mingchao Shao[2], Yunlong Guo [2]✉, Qingxin Tang [1]✉ & Yunqi Liu [2]✉

Developing highly active materials that efficiently utilize solar spectra is crucial for photocatalysis, but still remains a challenge. Here, we report a new donor-acceptor (D-A) covalent organic framework (COF) with a wide absorption range from 200 nm to 900 nm (ultraviolet-visible-near infrared light). We find that the thiophene functional group is accurately introduced into the electron acceptor units of TpDPP-Py (TpDPP: 5,5′-(2,5-bis(2-ethylhexyl)−3,6-dioxo-2,3,5,6-tetrahydropyrrolo [3,4-c]pyrrole-1,4-diyl)bis(thiophene-2-carbalde-hyde), Py: 1,3,6,8-tetrakis(4-aminophenyl)pyrene) COFs not only significantly extends its spectral absorption capacity but also endows them with two-photon and three-photon absorption effects, greatly enhancing the utilization rate of sunlight. The selective coupling of benzylamine as the target reactant is used to assess the photocatalytic activity of TpDPP-Py COFs, showing high photocatalytic conversion of 99% and selectivity of 98% in 20 min. Additionally, the TpDPP-Py COFs also exhibit the universality of photocatalytic selective coupling of other imine derivatives with ~100% conversion efficiency. Overall, this work brings a significant strategy for developing COFs with a wide absorption range to enhance photocatalytic activity.

The development of efficient photocatalysts is crucial for solar-driven photocatalysis benefit owing to its clean, eco-friendly, and renewable advantages[1-3]. Conventional inorganic photocatalysts, such as $TiO_2$[4,5], $ZnO$[6,7], and $CdS$[8,9] encounter constraints in modulating their broadband gaps, thereby limiting their capability to harness sunlight. Organic small-molecule photocatalyst, on the other hand, grapple with stability concerns, which complicates the processes of separation and recycling, impacting their overall sustainability[10]. Covalent organic frameworks (COFs), one of new crystalline porous materials, exhibiting enormous advantages, including high porosity[11], extended π conjugation[12,13], adjustable bandgaps[14], and good stability[15]. Significantly, the Pt-doped COFs reported by Lotch's group for continuous hydrogen production exhibit excellent structure and chemical stability, which has ignited extensive research on COFs-based photocatalysts[16]. Recently, COFs have been widely studied and applied in photocatalysts fields, including carbon dioxide reduction[17,18], photocatalytic hydrogen production[19,20], hydrogen peroxide synthesis[21,22], and organic synthesis[23,24]. The noteworthy

[1]Center for Advanced Optoelectronic Functional Materials Research, and Key Lab of UV-Emitting Materials and Technology of Ministry of Education, Northeast Normal University, 130024 Changchun, China. [2]Beijing National Laboratory for Molecular Sciences, Key Laboratory of Organic Solids, Institute of Chemistry Chinese Academy of Sciences, 100190 Beijing, China. [3]These authors contributed equally: Yuanding Fang, Youxing Liu. ✉e-mail: guoyunlong@iccas.ac.cn; tangqx@nenu.edu.cn; liuyq@iccas.ac.cn

**Fig. 1 | Design and synthesis of D-A COFs. a, b** Synthetic route of TpDPP and DPP organic molecules. **c, d** Molecular structure of TpDPP-Py COFs and DPP-Py COFs.

importance of imines as crucial intermediates in chemical synthesis, pharmaceuticals, and biology has prompted extensive investigations into the use of COFs for achieving efficient photocatalytic imine production[25-28]. In this pursuit, researchers have consistently focused on designing and synthesizing COFs with high photocatalytic active.

The common effective strategies for enhancing photocatalytic active are to promote the separation of photogenerated electron-hole, such as metal doping, heterojunctions, regulating band gap or the position of conduction and valence band[29]. Recently, the triazine-structured monomers[30,31], porphyrins[32,33], or phenolic functional group-containing monomers such as 2,4,6-triformyl resorcinol have been used to synthesize COFs for promoting the charge separation efficiency[27,34]. However, most COFs have a narrow absorption range typically between 200 nm and 550 nm, leading to reduced photocatalytic efficiency. To solve this bottleneck, developing novel strategies to expand the absorption range of COFs is key for enhancing their photocatalytic efficiency, but it still remains grand challenges.

Herein, we developed a novel strategy to precisely incorporate thiophene into the electron acceptor units of TpDPP-Py COFs (TpDPP: 5,5′-(2,5-bis(2-ethylhexyl)−3,6-dioxo-2,3,5,6-tetrahydropyrrolo [3,4-c] pyrrole-1,4-diyl)bis(thiophene-2-carbaldehyde), Py: 1,3,6,8-tetrakis(4-aminophenyl)pyrene), which extends its π-conjugated framework structure, significantly increasing the sunlight harvesting capabilities. In addition, the extended π-conjugated framework structure facilitates substantial interactions among non-adjacent chromophores without inducing aggregation, resulting in multi-photon absorption[35]. The fabricated TpDpp-Py COFs exhibit a wide absorption wavelength range of 200−900 nm and two/three-photon absorption effects, which enormous enhance the utilization rate of solar energy, thus exhibiting excellent performance of photocatalytic benzylamine (BA) coupling with conversion of 99% and selectivity of 98% in 20 min. In addition, the TpDPP-Py COFs also exhibit the universality of photocatalytic selective coupling of other imine derivatives with ~100% photocatalytic conversion efficiency. Overall, this work presents a significant strategy

for the development of COFs with a broad absorption range to enhance photocatalytic activity.

## Results and discussion

### Design and synthesis of TpDPP-Py COFs molecular with D-A structure

Diketopyrrolopyrrolopyrrole dialdehyde is widely recognized as a favorable candidate for the construction of donor–acceptor (D–A) structured COFs with enhanced light absorption performance owing to its well-established chromophore[36]. In the synthesis pathway, the thiophene functional groups with strong aromaticity and enhanced photocatalytic activity were selected as the side chain electron acceptor unit[37-39] (Fig. 1a, Supplementary Fig. 1–3). For comparison, the DPP (4,4′-(2,5-bis(2-ethylhexyl)−3,6-dioxo-2,3,5,6-tetrahydropyrrolo[3,4-c] pyrrole-1,4 diyl)dibenzaldehyde) molecule with phenyl groups as the side chain was also synthesized (Fig. 1b, Supplementary Fig. 4, 5). Further, the Py (Py: 1,3,6,8-tetrakis(4-aminophenyl)pyrene) molecular with abundant conjugated electrons was chosen as the electronic donor unit[40,41]. A novel TpDPP-Py COFs with large π-conjugated system was synthesized by employing TpDPP as the electron acceptor and Py as the electron donor (Fig. 1c). The reference DPP-Py COFs were synthesized by employing DPP as the electron acceptor and Py as the electron donor (Fig. 1d).

### Characterization of TpDPP-Py COFs

To confirm the chemical bonding properties, Fourier transform infrared (FT-IR) and Raman spectroscopy were carried out. The FT-IR spectroscopy showed the disappearance of the amino group of Py ($v_{N-H}$ = 3370–3346 cm$^{-1}$) and aldehyde group of TpDPP ($v_{HC=O}$ = 1693 cm$^{-1}$), accompanied by the presence of a new characteristic peak located at 1571 cm$^{-1}$ divided to C = N imine bonds, indicating the formation of TpDPP-Py COFs connected by imine bonds (Fig. 2a). Similarly, Raman spectroscopy displayed pronounced peaks at 1571 cm$^{-1}$, consistent with the presence of imine bonds observed in

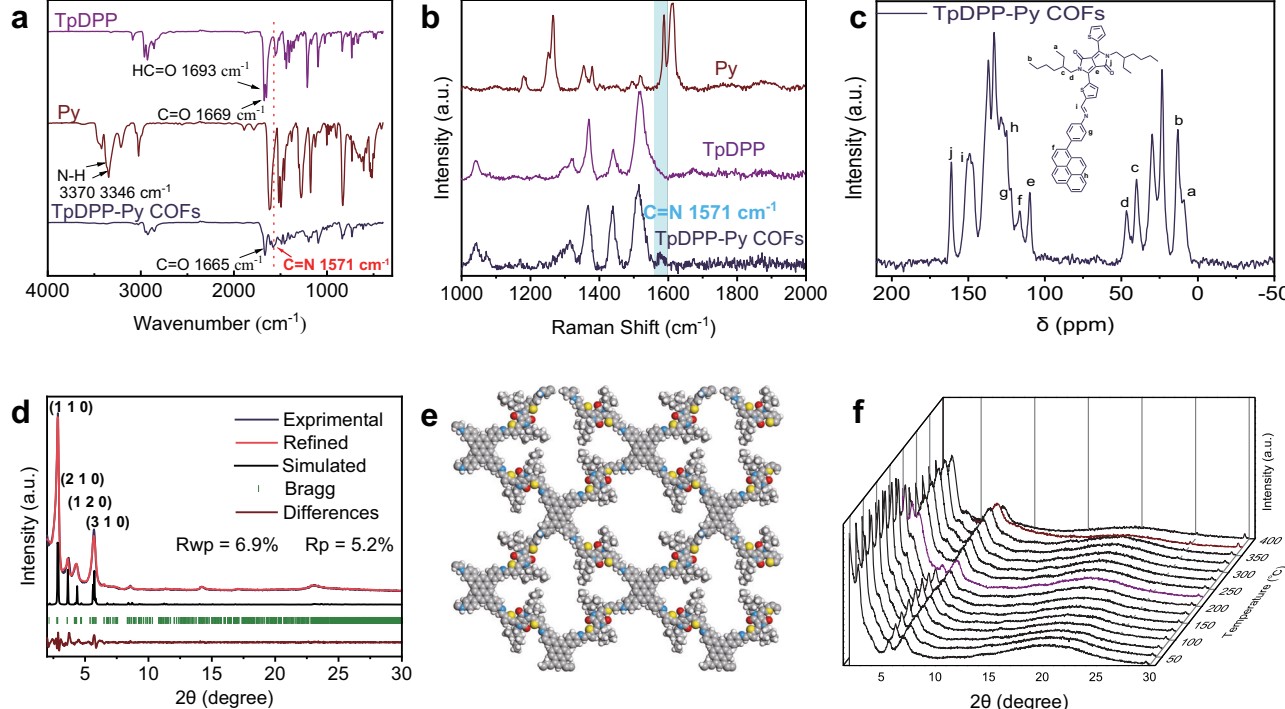

**Fig. 2 | Characterization of TpDPP-Py COFs. a** FT-IR spectra of TpDPP, Py and TpDPP-Py COFs. **b** Raman spectra of Py, TpDPP and TpDPP-Py COFs. **c** Solid-state $^{13}$C CP-MAS NMR spectrum of TpDPP-Py COFs. **d** Experimental and simulated XRD pattern of TpDPP-Py COFs (purple). **e** Simulated TpDPP-Py COFs molecular structure. **f** Temperature dependent XRD pattern of TpDPP-Py COFs.

the FT-IR analysis (Fig. 2b). X-ray Photoelectron Spectroscopy (XPS) data further confirmed the formation of imine bonds in TpDPP-Py COFs (Supplementary Fig. 6). Additionally, the solid-state $^{13}$C NMR (ss NMR) spectroscopy was performed to reveal the molecular framework structure of TpDPP-Py COFs, showing a chemical shift at 150 ppm corresponding to the imine bond (C = N) (Fig. 2c). In addition, the $^{15}$N ssNMR spectroscopy showed that the chemical shift located 162 ppm could be attributed to imine bond (Supplementary Fig. 7). These results provided additional evidence for the formation of COFs[42–47]. Overall, the above results provided powerful evidence for the forma-tion of imine bonded TpDPP-Py COFs. Further, The FT-IR spectra and $^{13}$C ssNMR also demonstrated the formation of DPP-Py COFs (Supple-mentary Fig. 8).

To further determine the ordered porous structure of TpDPP-Py COFs, N$_2$ adsorption and desorption curves, X-ray diffraction (XRD) analysis and Cryogenic transmission electron microscopy (cryo-TEM) were conducted. N$_2$ adsorption and desorption analysis revealed that the Brunauer-Emmett-Teller (BET) surface area and pore size of TpDPP-Py COFs were 408 m$^2$ g$^{-1}$ and 2.35 nm respectively, which was consistent with the theoretical simulations (Supplementary Fig. 9). The cell parameters of TpDPP-Py COFs and DPP-Py COFs are shown in Supplementary Tables 1, 2, respectively. The XRD pattern showed prominent peaks at 2.83°, 3.61°, 4.35° and 5.69°, corresponding to the crystal facets (110), (210), (120) and (310) respectively, with low-profile R-values of 5.2% and weighted-profile R-values of 6.9% (Fig. 2d, Sup-plementary Fig. 10). Compared to the A–B stacking arrangement observed in TpDPP-Py COFs (Supplementary Fig. 11), the A–A stacking arrangement aligns better with the experimental results (Fig. 2e). Additionally, temperature dependent XRD, Thermo-Gravimetric Ana-lysis (TGA) and Differential Scanning Calorimetry (DSC) results demonstrate the excellent structural stability of TpDPP-Py COFs (Fig. 2f, Supplementary Fig. 12). Furthermore, SEM and TEM images were used to study the surface morphology of the TpDPP-Py COFs and DPP-Py COFs (Supplementary Fig. 13, Fig. 14). TEM images revealed the nanosheet morphology of TpDPP-Py COFs, displaying an average

particle size of 650 nm (Supplementary Fig. 14). Additionally, cryo-TEM provided in-depth insights into the microstructural and lattice structure of the COF material (Fig. 3a). The crystal spacing of 3.11 nm was corresponded to the (110) crystal plane (Fig. 3b). The Fast Fourier Transform (FFT) patterns also exhibited a tetragonal structure, which was a match to the simulated molecular structure (Fig. 3c). Elemental mapping showed the uniform distribution of sulfur (S), oxygen (O), nitrogen (N), and carbon (C) on TpDPP-Py COFs (Fig. 3d). Thus, the above results confirmed the successful synthesis of TpDPP-Py COFs with ordered porous structure.

## Optical characteristics and DFT theoretical calculations of TpDPP-Py COFs

The optical band gap of TpDPP-Py COFs was calculated as 1.38 eV through the equation $E_g$ (eV) = 1240 / $\lambda$ (nm) (Fig. 4a). The UV-vis spectrum showed that TpDPP-Py COFs exhibited broad absorption peaks range from 200 to 900 nm, indicating a narrower optical bandgap and higher solar energy utilization efficiency compared to DPP-Py COFs (Fig. 4b, Supplementary Fig. 15). The photoluminescence fluorescence spectrum exhibited that TpDPP-Py COFs could be excited within the wavelength range from 300 nm to 700 nm, with an emission peak observed at 810 nm, signifying favorable absorption properties in the visible and near-infrared regions (Fig. 4c, Supplementary Fig. 16). The fluorescence decay of TpDPP-Py COFs exhibited a dual exponen-tial decay model, with fluorescence decay times of 0.0595 ns and 0.299 ns (Fig. 4d, Supplementary Fig. 17), which were shorter than DPP-Py COFs (0.168 ns and 0.862 ns) (Supplementary Fig. 18), indicating more efficient charge separation efficiency and transport dynamics[48,49].

The HOMO and LUMO energy levels of TpDPP Py COFs were cal-culated to be −4.67 eV and −3.39 eV, respectively (Fig. 4e), yielding a theoretical band gap of 1.28 eV. Moreover, compared with DPP-Py COFs, TpDPP-Py COFs exhibited excellent two-photon and three-photon absorption characteristics (Fig. 4f, g, Supplementary Fig. 19), implying more effectively utilization of sunlight. Figure 4h shows the photocurrent

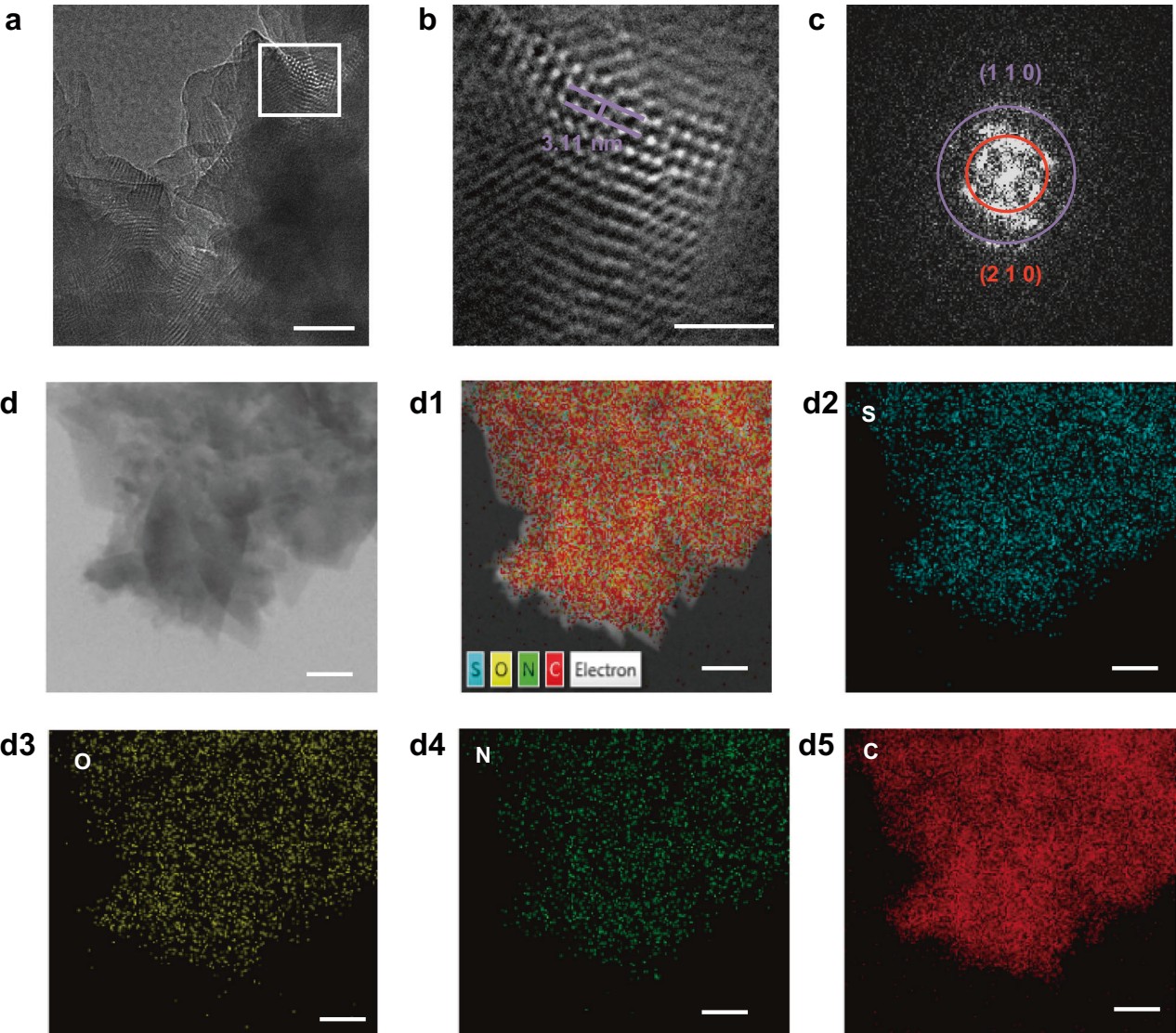

**Fig. 3 | TEM characterization of TpDPP-Py COFs. a** Cryo-TEM images of TpDPP-Py COFs (scale bar, 50 nm). **b** High resolution Cryo-TEM of the TpDPP-Py COFs (scale bar, 20 nm). **c** FFT patterns of TpDPP-Py COFs. **d** HAADF-STEM image and the corresponding STEM-EDS elemental mapping images of TpDPP-Py COFs (scale bar, 100 nm) (d1, merge; d2, S; d3, O; d4, N; d5, C).

response curves of TpDPP-Py COFs and DPP-Py COFs. The TpDPP-Py COFs exhibited a photocurrent density of 0.054 µA cm$^{-2}$, which was significantly higher than that of DPP-Py COFs (0.035 µA cm$^{-2}$). The high photocurrent density of TpDPP-Py COFs mainly benefits from its excellent light absorption characteristics.

### Photocatalytic benzylamine (BA) coupling and reaction mechanism

To assess the photocatalytic properties, TpDPP-Py COFs were used for the selective photocatalytic coupling of benzylamine (BA) to N-benzylidenebenzylamine. The gas chromatogram (GC) was carried out to detect the conversion rate of N-benzylidenebenzylamine (Supplementary Fig. 20, 21). Figure 5a shows the activity and selectivity of converting BA to N-benzylidenebenzylamine over TpDPP-Py COFs. The TpDPP-Py COFs exhibit a high photocatalytic coupling conversion of BA of 99% and high selectivity of 98% in 20 min, which is significantly better than DPP-Py COFs (Fig. 5b). Meanwhile, the performance of TpDPP-Py COFs photocatalytic conversion of BA to N-benzylidenebenzylamine is also better than most reported photocatalysts (Fig. 5c, Supplementary Table 3). Figure 5d shows the activity and selectivity of different batches TpDPP-Py COFs for photocatalytic conversion of BA to

N-benzylidenebenzylamine. No obvious attenuation indicates the excellent performance stability of TpDPP-Py COFs. Furthermore, XRD and FT-IR spectra confirm that the crystal structure of TpDPP-Py COFs does not undergo significant changes after the photocatalytic reaction (Supplementary Fig. 22), validating its excellent structural stability.

Understanding the reaction mechanism was very necessary for both fundamental research and application research. We found that BA could not be converted to N-benzylidenebenzylamine without photocatalyst or dark conditions (Supplementary Table 4), implying the necessary of photocatalyst and light irradiation for the selective photocatalytic coupling of BA. Besides, oxygen also played an important role in the selective photocatalytic coupling of BA (Supplementary Table 4). Thus, another important problem to be solved is how oxygen plays a role in selective photocatalysis coupling of BA. Electron paramagnetic resonance (EPR) was carried out to study the active intermediates of O$_2$. Figure 5e shows that the EPR detection of the $^1$O$_2$ can be detected under light irradiation, while it disappears under dark condition, confirming that the triplet state oxygen is converted into singlet oxygen under light irradiation. The $^1$O$_2$ intensity over TpDPP-Py COFs is higher than that of DPP-Py COFs (Fig. 5f and Supplementary Fig. 23), implying its higher efficiency of oxygen conversion, which is

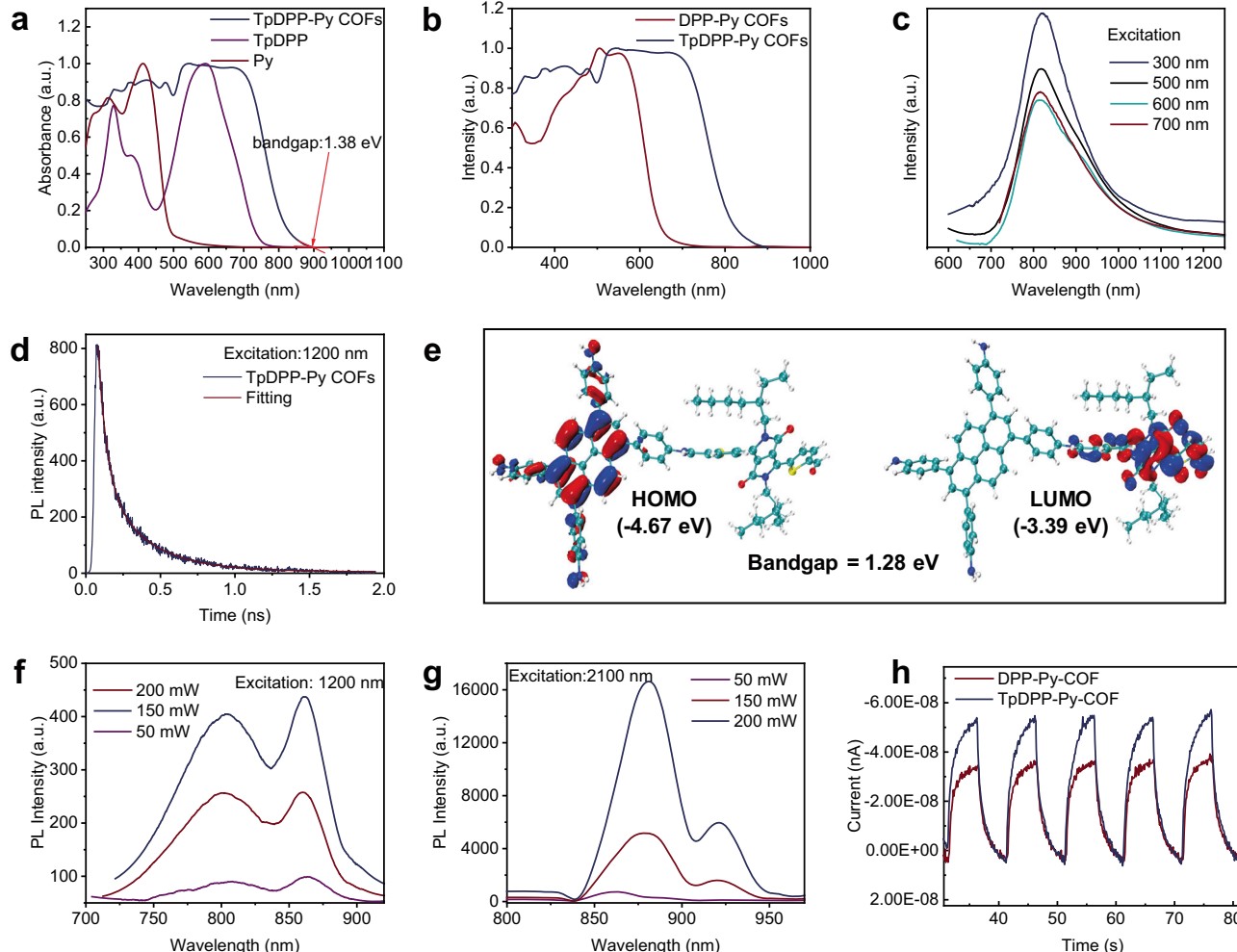

**Fig. 4 | Optical characteristics and DFT theoretical calculations of TpDPP-Py COFs. a** Kubelka-Munk function analysis of UV/Vis absorption spectra for TpDPP-Py COFs, TpDPP, and Py precursor materials, along with diffuse reflection spectra. **b** Kubelka-Munk Function UV/Vis Absorption Spectra of TpDPP-Py COFs and DPP-Py COFs. **c** Solid-state fluorescence spectra of TpDPP-Py COFs under different excitation wavelengths. **d** Time-resolved photoluminescence spectra of TpDPP-Py COFs ($\lambda_{ex} = 1200$ nm). **e** Molecular orbital amplitude plots of HOMO and LUMO of the ligand and *p*-extended ligand calculated at the ωb97xd/6-31 G* basis set. **f** Two-photon induced solid-state fluorescence emission of TpDPP-Py COFs at different units ($\lambda_{ex} = 1200$ nm). **g** three-photon induced solid-state fluorescence emission of TpDPP-Py COFs at different units ($\lambda_{ex} = 2100$ nm). **h** transient photocurrent responses of the TpDPP-Py COFs and DPP-Py COFs.

mainly benefited from the high utilization of sunlight of TpDPP-Py COFs. Further, Fig. 5g shows the EPR signal of the $\cdot O_2^-$ anion radical over TpDPP-Py COFs. No obvious signal can be detected under dark condition, while EPR signal of $\cdot O_2^-$ anion radical is obtained under light irradiation, confirming that $^1O_2$ can capture the photogenerated electrons to form $\cdot O_2^-$ anion radical. The EPR intensity of $\cdot O_2^-$ anion radical over TpDPP-Py COFs is higher than that of DPP-Py COFs (Fig. 5h and Supplementary Fig. 24), indicating that more $^1O_2$ are converted into $\cdot O_2^-$ anion radical on TpDPP-Py COFs. In addition, the photogenerated hole intensity of TpDPP-Py COFs with the presence of BA is significantly lower than that without BA (Fig. 5i), indicating that BA consumes photogenerated holes in the system.

Based on the above discussion, the mechanism of selective photocatalytic coupling of BA to generate N-benzylidenebenzylamine can be explained (Supplementary Fig. 25). Firstly, TpDPP-Py COFs are excited to produce photogenerated electron-hole pairs. And then, the $^1O_2$ captures photogenerated electrons and is reduced to $\cdot O_2^-$ anion radical. Meanwhile, the α-carbon of BA is activated by the photogenerated hole. Further, the $\cdot O_2^-$ anion radical is used to react with activated BA molecule to generate benzaldehyde intermediate. Finally, the unstable benzaldehyde intermediates rapidly react with BA to synthesize N-benzylidenebenzylamine (Supplementary Fig. 25).

## Study on the universality of photocatalytic amine coupling

The TpDPP-Py COFs photocatalyst was also used to explore selective photocatalytic coupling of amine derivatives (2-ethoxyaniline, 3-ethoxyaniline, 4-ethoxyaniline, 2-thiophene methylamine, 4-fluoroaniline, 4-chlorobenzylamine) under the visible light irradiation. As shown in Fig. 6, with an increase in reaction time, the conversion of all amine derivatives increases and then stabilizes, indicating complete conversion of the amine derivatives and confirming the universality of TpDPP-Py COFs for photocatalytic amine coupling. Furthermore, the yield of all amine conversion to corresponding coupling products is higher than 97%, and the conversion efficiency of 2-thiophene methylamine approaching 100% within 1 h (Fig. 6c), which indicates the high photocatalytic activity of TpDPP-Py COFs. The excellent photocatalytic performance is benefited from the wide absorption properties and multi-photon absorption effect.

In summary, we present a novel TpDPP-Py COFs for the first time by integrating thiophene into DPP organic ligands, resulting in a broad absorption range spanning from 200 nm to 900 nm (ultraviolet-visible-near infrared light). Our study illustrates that the extended π-conjugated framework structure not only enhances the absorption capacity but also confers multi-photon absorption effects, significantly improving sunlight utilization and photocatalytic activity. Utilizing the

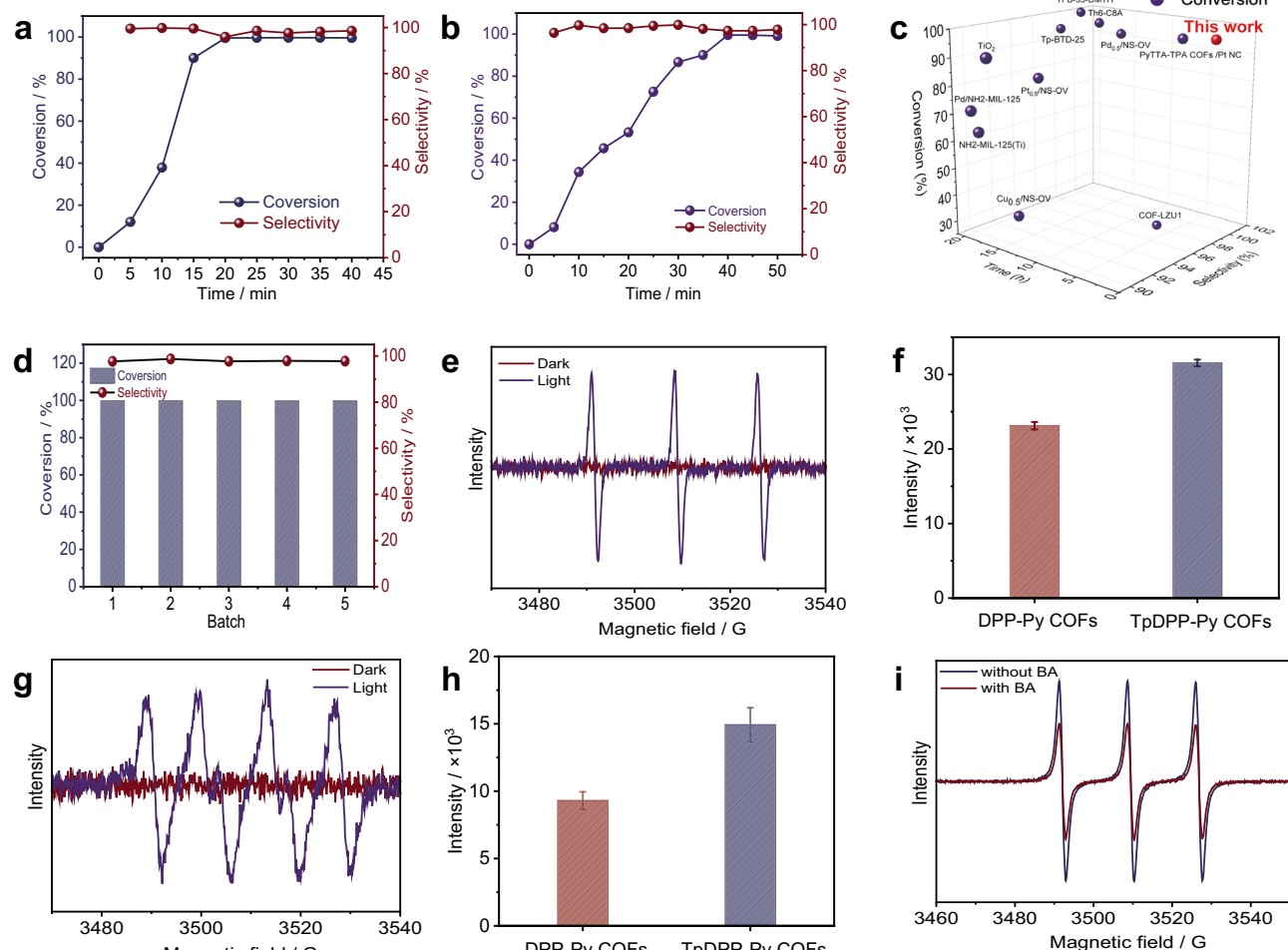

**Fig. 5 | TPDPP-Py COFs photocatalytic BA coupling and reaction mechanism.**
**a** Activity and selectivity of TPDPP-Py COFs photocatalytic conversion of BA to N-benzylidenebenzylamine. **b** Activity and selectivity of DPP-Py COFs photocatalytic conversion of BA to N-benzylidenebenzylamine. **c** performance comparison of TpDPP-Py COFs with reported photocatalyst. **d** Repeatability measurement of TpDPP-Py COFs photocatalytic conversion of BA to N-benzylidenebenzylamine.
**e** EPR detection of the $^1O_2$ trapped by TEMP over TpDPP-Py COFs under dark and light irradiation. **f** Comparison of EPR intensity of $^1O_2$ between TpDPP-Py COFs and DPP-Py COFs. **g** EPR detection of the $\cdot O_2^-$ anion radical trapped by DMPO over TpDPP-Py COFs under dark and light irradiation. **h** Comparison of EPR intensity of $\cdot O_2^-$ anion radical between TpDPP-Py COFs and DPP-Py COFs. **i** EPR detection of the hole over TpDPP-Py COFs with and without BA.

TpDPP-Py COFs for the selective photocatalytic coupling of benzylamine (BA) resulted in a high BA coupling conversion rate of 99% and a selectivity of 98% within a 20-min timeframe, surpassing that of most reported active materials. Furthermore, the TpDPP-Py COFs demonstrate a general applicability for the photocatalytic coupling of various amine derivatives. This work introduces a new strategy for the development of innovative COFs photocatalysts, which is of great significance for promoting the practical application of COFs in the field of photocatalysis.

## Methods
### Synthesis of TpDPP-Py COFs
Synthesis of TpDPP-Py-COFs: 5,5′-(2,5-bis(2-ethylhexyl)−3,6-dioxo-2,3,5,6-tetrahydropyrrolo [3,4-c]pyrrole-1,4-diyl)bis(thiophene-2-carbaldehyde) (TpDpp) (12 mg, 0.02 mmol), and 1,3,6,8-tetrakis(4-aminophenyl)pyrene (Py) (6 mg, 0.01 mmol) were weighed into a 10 mL glass tube. Afterwards, o-DCB: mesitylene = 4:1 = 1.0 mL was added and the mixture was sonicated for 5 min. After addition of 0.1 mL AcOH (6 M, aqueous), the tube was degassed by the three freeze-pump-thaw cycles and sealed with flame. Upon warming to room temperature, the sealed tube was heated at 120 °C for 72 h. The formed precipitate was collected by filtration, washed with methanol and tetrahydrofuran (THF) and further activated by Soxhlet extraction using methanol and

THF for 72 h. Finally, the solid was collected and dried at 120 °C for 12 h under vacuum to give TpDPP-Py COFs as the dark purple powder (15 mg, 60% yield).

### Synthesis of DPP-Py COFs
Synthesis of DPP-Py COFs: 4,4′-(2,5-bis(2-ethylhexyl)−3,6-dioxo-2,3,5,6-tetrahydropyrrolo[3,4-c]pyrrole-1,4-diyl)dibenzaldehyde (DPP) (12 mg, 0.02 mmol), and1,3,6,8-tetrakis(4-aminophenyl)pyrene (Py) (6 mg, 0.01 mmol) were weighed into a 10 mL glass tube. Afterwards, 1.4-dioxane: mesitylene = 1:1 = 1.0 mL was added and the mixture was sonicated for 5 min. After addition of 0.1 mL AcOH (6 M, aqueous), the tube was degassed by the three freeze-pump-thaw cycles and sealed with flame. Upon warming to room temperature, the sealed tube was heated at 120 °C for 72 h. The formed precipitate was collected by filtration, washed with methanol and tetrahydrofuran (THF) and further activated by Soxhlet extraction using methanol and THF for 72 h. Finally, the solid was collected and dried at 120 °C for 12 h under vacuum to give DPP-Py COFs as shiny red powder (10 mg, 50% yield). The synthesis route of TpDPP and DPP is in Supplementary Fig. S1−5.

### Selective photocatalysis coupling of benzylamine (BA)
5 mg photocatalyst was added into acetonitrile solution containing 1% benzylamine in the Pyrex vessel reactor. the mixed solution was

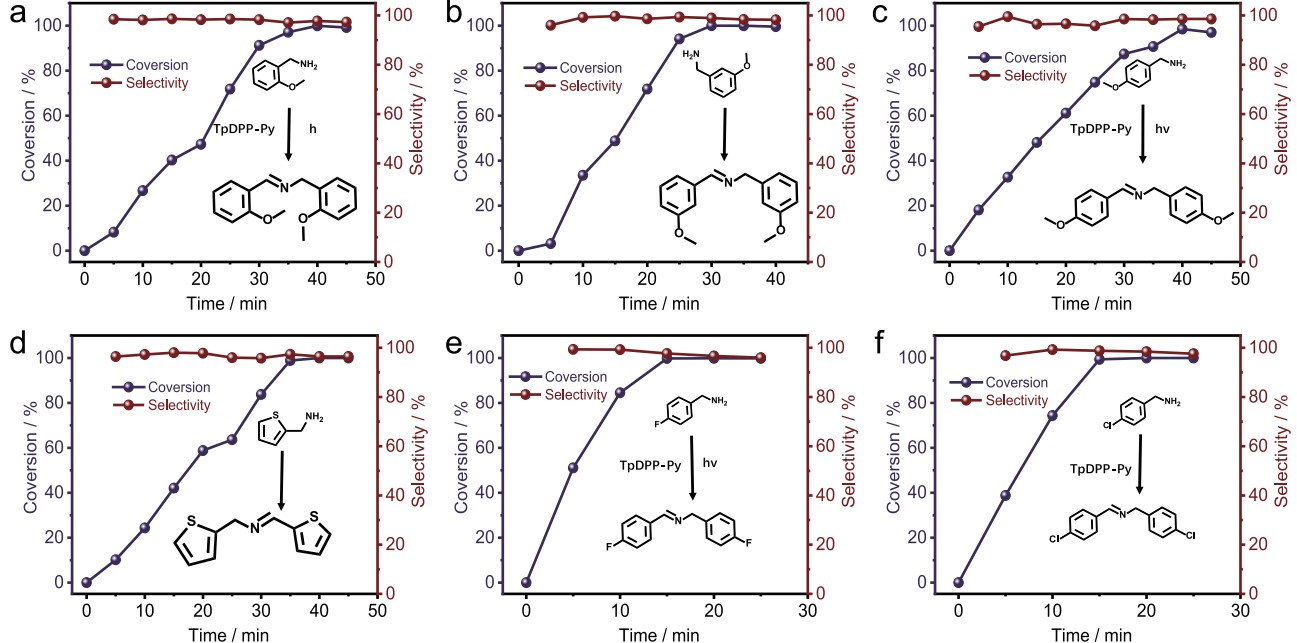

**Fig. 6 | Activity and selectivity of TpDPP-Py COFs photocatalytic conversion of different amines derivatives. a** 2-ethoxyaniline, **b** 3-ethoxyaniline, **c** 4-ethoxyaniline, **d** 2-thiophene methylamine, **e** 4-fluoroaniline, and **f** 4-Chlorobenzylamine.

bubbled with oxygen for 30 min before light irradiation. And then, the suspension solution was irradiated by a 300 W Xe lamp (Micro-solar300, Beijing Perfectlight) equipped with AM1.5 G solar intensity. The gas chromatography (GC-7890B) with a flame ionization detector (FID) detector was carried out to detect the conversion rate of N-benzylidenebenzylamine. The photocatalytic coupling of amine derivatives (2-ethoxyaniline, 3-ethoxyaniline, 4-ethoxyaniline, 2-thiophene methylamine, 4-fluoroaniline, 4-chlorobenzylamine) is similar to that of BA.

### Characterization

Powder X-ray diffraction (PXRD) patterns were recorded on PANalytical Empyrean diffractometer for Cu/Kα radiation (λ = 1.5416 nm) The samples were spread on the square recess of XRD sample holder as a thin layer. Variable temperature PXRD was recorded at a rate of 10 °C min$^{-1}$ and held for 2 min. Themis 300 (cryo-TEM) at an accelerating voltage of 200 KV. Imaging was conducted using low-dose techniques and the dose rate was <5.0 e/Å$^2$/s. Disperse 2 mg of COF into 1 ml of ethanol and add dropwise in the microgrid. Nanosheets were observed at JEM-2100 (200KV). Scanning Electron Microscopy (SEM) images were recorded using a Zeiss Gemini 300 scanning electron microscopy. Infrared spectra were measured on Lambda 1050+ under vacuum. All spectra were background corrected. Thermogravimetric analysis (TGA) measurements were carried out on a PerkinElmer series 7 thermal analysis system under N$_2$ at a heating rate of 10 °C min$^{-1}$. $^{13}$C cross polarization magic angle spinning nuclear magnetic resonance ($^{13}$C CP/MAS NMR) spectra was recorded on a Bruker NEO 600 WB spectrometer. Samples were packed in 4 mm ZrO$_2$ rotors, which were spun at 8 kHz in a double resonance MAS probe. All spectra were background corrected. Absorption spectra were recorded by PerkinElmer Lambda 1050 + UV-vis-NIR. UV-vis spectrophotometer at room temperature. The absorption spectra of solid powder employed integrating sphere mode with Ba$_2$SO$_4$ background. Fluorescence spectra of COF powders was using fluorescence spectrometer FLS1000, and the transient photoluminescence decay lifetime profiles of COF powder was using Astrella.

### Data availability

The authors declare that the experimental data supporting the results of this study can be found in the paper and its Supplementary Information file. The experimental results of the study along with other simulation data are provided in the Supplementary Information file. The detailed simulation files for the study are available from the corresponding author upon request. Source data are provided with this paper.

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

## Acknowledgements

This work was financially supported by the Strategic Priority Research Program of CAS (XDB0520101), the National Natural Science Foundation of China (U22A6002, 22173109, 52303363), the National Key R&D Program of China (2018YFA0703200), the CAS Project for Young Scientists in Basic Research (YSBR-053), the CAS-Croucher Scheme for Joint Laboratories, Lu Jiaxi international team (GJTD-2020-02), the CAS Cooperation Project (121111KYSB20200036), and the Beijing Nova Program (20220484173). The authors gratefully acknowledge the

assistance of Ningning Wu in data collection for nuclear magnetic analysis. We also extend our appreciation to Jiling Yue and Kaiang Liu for their valuable contributions in characterizing the cryo-TEM experiments. Additionally, we would like to thank Meirong Liu for her help in operating the femtosecond laser spectrometer.

## Author contributions

Y.F. and Yo.L contributed equally to this work. Y.G., Q.T. and Yu.L proposed and supervised the project. Y.F., Yo.L conceived the idea and designed the experiments. H.H. conducted DFT calculations. J.S. conducted Raman spectroscopy characterization. J.H. conducted transient photocurrent response. W.G. and M.S. assisted in the synthesis of materials. Q.T., F.Z., X.W. and Y.G. assisted in the writing of the article. X.W. assisted in the spectral experiment. Y.F. and Yo.L wrote the manuscript and all authors reviewed it.

## Competing interests

The authors declare no competing interests.
