## [Peer Review File · Nature Communications]

Design and Synthesis of Broadband Absorption Covalent Organic Framework for Efficient Artificial Photocatalytic Amine CouplingREVIEWER COMMENTS

Reviewer #1 (Remarks to the Author):

The paper presents a detailed study on designing, synthesizing, and applying a novel donor-acceptor (D-A) covalent organic framework (COF) exhibiting broad-band light absorption and high photocatalytic performance for amine coupling.

The solid-state ^{13}C NMR spectroscopy was performed, revealing chemical shifts that indicate the formation of the COFs, with a specific mention of a shift at 150 ppm for the imine bond. However, extending the NMR scan range starting from 200 ppm could provide a more comprehensive understanding of the COF structure, possibly identifying additional functional groups or verifying the uniformity of the framework [J. Am. Chem. Soc. 2023, 145, 43, 23802–23813; J. Am. Chem. Soc. 2023, 145, 34, 18855–18864].

While the paper includes solid-state NMR data, a more detailed solid-state NMR (SSNMR) analysis, possibly including ^{15}N NMR spectroscopy, could offer deeper insights into the framework's architecture, cross-linking, and the distribution of functional groups, which are critical for understanding the material's photocatalytic efficiency and stability [J. Am. Chem. Soc. 2023, 145, 26, 14475–14483].

The article lacks explicit information on particle size, which is a significant oversight given the particle size's impact on photocatalytic activity, light absorption, and material dispersion in applications. Including particle size distribution data, possibly obtained through Dynamic Light Scattering (DLS) or Transmission Electron Microscopy (TEM), would enhance the material characterization [J. Am. Chem. Soc. 2023, 145, 3, 1649–1659].

The paper successfully introduces a promising COF material with significant photocatalytic activity and broad-band light absorption. However, including the suggested NMR scan range, a detailed SSNMR analysis, and particle size data would comprehensively characterize the COF's structure and properties, offering a fuller understanding of its performance and potential applications.

Reviewer #2 (Remarks to the Author):

The enhancement of solar utilization has been acknowledged as a longstanding breakthrough in the field of photocatalysis. Thus, the design and synthesis of photocatalysts with broad absorption capabilities has garnered growing attention. In this manuscript, the authors have developed a novel D-A covalent organic framework by incorporating thiophene into the DPP organic ligands. The resulting TpDPP-Py COFs display a broad absorption spectrum from ultraviolet to near-infrared regions. Additionally, TpDPP-Py COFs exhibit exceptional two-photon and three-photon absorption properties, enhancing the utilization efficiency of sunlight and photocatalytic activity. Overall, This study presents a novel methodology for the design of broadband absorption COFs, demonstrating exceptional photocatalytic oxidation performance in the oxidative coupling of benzylamine under white light irradiation. However, there still are some questions and problems need to be revised in the manuscript before it could be published.

1. In the abstract, the term "conversion rate" may not be appropriate in line 29.
2. In the introduction section, the authors have emphasized the NLO properties of COFs. Is there any relationship between NLO response and photocatalytic activity?
3. In lines 88-89, the authors indicate that a TpDpp unit with higher electron deficiency was synthesized. Additional evidence was required to support this statement.
4. The crystal facets in lines 137-138 do not align with the findings presented in Figure 2d and Figure 3c.
5. The inclusion of the low-profile R-value (R_p) and weighted-profile R-value (R_{wp}) in Figure 2d is essential to ensure the precision of the refinement outcomes.
6. In the section titled "TEM characterization of TpDPP-Py COFs," the authors present findings derived from PXRD, BET, and Raman measurements. The authors should carefully consider the alignment between the title's scope and their discussion's logical progression.

7. The representation of "BET: 408 m² g⁻¹" suggested to be changed to "surface area: 408 m² g⁻¹" in Figure S8 of the Supporting Information.
8. The XPS peak-differentiating and imitating patterns of TpDPP-Py COFs were not rational.
9. The lattice parameters may not suitable for the symmetry of TpDPP-Py COFs.
10. The authors are required to replace the Chinese text in Figure S7a Figure S9b with English.
11. Please optimize the layout of Table S1 in the Supporting Information for improved readability.
12. The author mentions that "it is the best performance among the reported materials" in page 1 line 27, however, in Table S1 of the Supporting Information, reference 10 has already achieved 99% conversion and 100% selectivity. The author needs to reconsider their statement.
13. In the line 77, the authors state that TpDPP-Py COFs exhibited the best catalytic performance compared with other reported materials. It might be overly definitive and exaggerate in nature.
14. In Figure 4a, the authors have indicated that the bandgap of TpDPP-Py COF is 1.38 eV. It is necessary to provide the calculation formula, as observed in Figure 4e and f, the calculated bandgap is 1.28 eV, with a 0.1 eV error between the calculated value and the actual value.
15. The stability of the catalyst is also an important standard for evaluating performance. The authors should supplement the characterization results such as infrared and XRD after the catalyst has been cycled, to prove that the catalyst has not undergone significant changes.
16. The authors need to consider the clarity and aesthetics of the figures.

Reviewer #3 (Remarks to the Author):

In this manuscript, Fang et al. developed a new COFs with a wide absorption range, which significantly promotes the sunlight utilization efficiency for the artificial photocatalytic amine coupling. The thiophene functional groups was introduced into the donor-acceptor units for the synthesis of new TpDPP-Py COFs, exhibiting a wide absorption range from 200 nm to 900 nm (ultraviolet-visible-near infrared light) and two-photon fluorescence emission as well as Three-photon fluorescence emission, which is a major accomplishment for COFs molecular design and photocatalysis field. The photocatalytic performance with ~ 100% photocatalytic conversion rate is exciting. Overall, the results reported in this manuscript are quite interesting and suitable for publication in Nature Communications after minor revisions.

Specific comments:

1. Donor and acceptor unit is important for the absorption of COFs. How to judge the ability of acceptor is not clear statement. For example, in figure 1c and 1d, strong or weak acceptor might be not appropriate. It is better to delete this. If you want to compare it, it is better to comparing the LUMO levels of two different acceptors.
2. Optoelectronic transmission dynamics data are missing in the article, the time-resolved photoluminescence spectra of TpDPP-Py COFs are suggested to supplement them.
3. The importance of thiophene functional groups was proposed in this study, but there is a lack of complete supporting data.
4. The author claims that the crystal structure of TpDPP-Py COFs is A-A stacking, as the comparison, the A-B stacking structure of simulated COFs should be provided and further sufficient explained.
5. The reason for the high photocatalytic efficiency of TpDPP Py COFs as short-lived materials should be further explained in detail.
6. The mechanism of artificial photocatalytic amine coupling is also important, the author should provide a detailed explanation.
7. The authors should explain why "Artificial" is added in the title of the manuscript.
8. It is recommended to correct some minor errors in spelling, grammar, format, etc., to make the article more professional and readable.

A point-by-point response to the reviewers' comments

Responses (R) to the Comments (C)

Responses to the reviewer 1:

The paper presents a detailed study on designing, synthesizing, and applying a novel donor-acceptor (D-A) covalent organic framework (COF) exhibiting broad-band light absorption and high photocatalytic performance for amine coupling.

Answer: Thank you very much for your very positive comments. We have revised our paper according to your suggestions.

C1: The solid-state ^{13}C NMR spectroscopy was performed, revealing chemical shifts that indicate the formation of the COFs, with a specific mention of a shift at 150 ppm for the imine bond. However, extending the NMR scan range starting from 200 ppm could provide a more comprehensive understanding of the COF structure, possibly identifying additional functional groups or verifying the uniformity of the framework [J. Am. Chem. Soc. 2023, 145, 43, 23802–23813; J. Am. Chem. Soc. 2023, 145, 34, 18855–18864].

R1: Thank you for your suggestion. We have extended the NMR scan range to conduct a more comprehensive analysis. The COF structure was further elucidated and analyzed in **Fig. R1**. The related references (J. Am. Chem. Soc. 2023, **145**, 43, 23802–23813; J. Am. Chem. Soc. 2023, **145**, 34, 18855–18864) have been cited in our manuscript. The revised manuscript has been highlighted in red on Page 4, Line 18, and **Fig. 2c**.

Fig. R1. Solid-state ^{13}C CP-MAS NMR spectrum of TpDPP-Py COFs.

C2: While the paper includes solid-state NMR data, a more detailed solid-state NMR (SSNMR) analysis, possibly including ^{15}N NMR spectroscopy, could offer deeper insights into the framework's architecture, cross-linking, and the distribution of functional groups, which are critical for understanding the material's photocatalytic efficiency and stability [J. Am. Chem. Soc. 2023, 145, 26, 14475–14483].

R2: Thank you for your suggestion. The ^{15}N NMR spectroscopy of TpDPP-Py COFs has been provided, and further detailed solid-state NMR analysis is presented in **Fig. R2**. The related reference (J. Am. Chem. Soc. 2023, **145**, 26, 14475–14483) has been

cited in our manuscript. The revised manuscript has been highlighted in red on Page 4, Line 18-19, and **Supplementary Fig. 7**.

Fig. R2. ^{15}N ssNMR spectroscopy of TpDPP-Py COFs and TpDPP.

C3: The article lacks explicit information on particle size, which is a significant oversight given the particle size's impact on photocatalytic activity, light absorption, and material dispersion in applications. Including particle size distribution data, possibly obtained through Dynamic Light Scattering (DLS) or Transmission Electron Microscopy (TEM), would enhance the material characterization [J. Am. Chem. Soc. 2023, 145, 3, 1649–1659].

R3: Thank you for your suggestion. TEM was used to assess the particle size of TpDPP-Py COFs, revealing an average size of 650 nm as shown in **Fig. R3**. The revised manuscript has been highlighted in red on Page 6, Line 9, and **Supplementary Fig. 14**.

Fig. R3. THE HR-TEM image of TpDPP-Py COFs.

C4: The paper successfully introduces a promising COF material with significant photocatalytic activity and broad-band light absorption. However, including the suggested NMR scan range, a detailed SSNMR analysis, and particle size data would comprehensively characterize the COF's structure and properties, offering a fuller understanding of its performance and potential applications.

R4: Thank you for your comment. The NMR analysis, detailed SSNMR analysis, and particle size data have been provided in **Fig. R1**, **Fig. R2**, and **Fig. R3**, respectively. The revised manuscript has been marked in red on Page 4, Line 18, and **Fig. 2c**, Page 4, Line 18-19, **Supplementary Fig. 7**, and Page 6, Line 9, and **Supplementary Fig. 14**.

Figure R1. Solid-state ^{13}C CP-MAS NMR spectrum of TpDPP-Py COFs.

Fig. R2. ^{15}N ssNMR spectroscopy of TpDPP-Py COFs and TpDPP.

Fig. R3. THE HR-TEM image of TpDPP-Py COFs.

Responses to the reviewer 2:

The enhancement of solar utilization has been acknowledged as a longstanding breakthrough in the field of photocatalysis. Thus, the design and synthesis of photocatalysts with broad absorption capabilities has garnered growing attention. In this manuscript, the authors have developed a novel D-A covalent organic framework by incorporating thiophene into the DPP organic ligands. The resulting TpDPP-Py COFs display a broad absorption spectrum from ultraviolet to near-infrared regions. Additionally, TpDPP-Py COFs exhibit exceptional two-photon and three-photon absorption properties, enhancing the utilization efficiency of sunlight and photocatalytic activity. Overall, this study presents a novel methodology for the design of broadband absorption COFs, demonstrating exceptional photocatalytic oxidation performance in the oxidative coupling of benzylamine under white light irradiation. However, there still are some questions and problems need to be revised in the manuscript before it could be published.

Answer: Thank you very much for your very positive comments. We have revised our paper according to your suggestions.

C1: In the abstract, the term "conversion rate" may not be appropriate in line 29.

R1: Thank you for your comment. We modify "conversion rate" to "conversion efficiency". The revised section is marked in red color on page 1, line 26.

C2: In the introduction section, the authors have emphasized the NLO properties of COFs. Is there any relationship between NLO response and photocatalytic activity?

R2: Thank you for your comment. The NLO is the physical optical properties of COFs, which is not necessarily related to photocatalytic activity. We have deleted this sentence.

C3: In lines 88-89, the authors indicate that a TpDpp unit with higher electron deficiency was synthesized. Additional evidence was required to support this statement.

R3: Thank you for your suggestion. We have revised and reorganized this sentence as " In the synthesis pathway (**Fig. 1a**), the thiophene functional group exhibits strong aromaticity and enhances photocatalytic activity, making it an ideal choice as the side chain electron acceptor unit ". The revised section is marked in red color on page 3, line 21-22.

C4: The crystal facets in lines 137-138 do not align with the findings presented in Figure 2d and Figure 3c.

R4: Thank you for your comment. We have rechecked and revised the manuscript. The revised section is marked in red color on page 5, line 18, 19.

C5: The inclusion of the low-profile R-value (R_p) and weighted-profile R-value (R_{wp}) in Figure 2d is essential to ensure the precision of the refinement outcomes.

R5: Thank you for your suggestion. We have incorporated the low-profile R-value (R_p) of 5.2% and the weighted-profile R-value (R_{wp}) of 6.9% in **Fig. R4**. These values are

displayed in the revised manuscript in **Fig. 2d** on page 5, line 20, highlighted in red.

Fig. R4. Experimental and simulated XRD pattern of TpDPP-Py COFs (purple).

C6: In the section titled "TEM characterization of TpDPP-Py COFs," the authors present findings derived from PXRD, BET, and Raman measurements. The authors should carefully consider the alignment between the title's scope and their discussion's logical progression.

R6: Thank you for your comment. We have re-adjusted the section title and checked logical progression.

C7: The representation of "BET: 408 m² g⁻¹" suggested to be changed to "surface area: 408 m² g⁻¹" in Figure S9 of the Supporting Information.

R7: Thank you for your suggestion. The "BET: 408 m² g⁻¹" has been revised to "surface area: 408 m² g⁻¹" in **Fig. R5**, and the revised manuscript is presented in **Supplementary Fig. 9**.

Fig. R5. (a) Nitrogen sorption isotherm of TpDPP-Py COFs.

C8: The XPS peak-differentiating and imitating patterns of TpDPP-Py COFs were not rational.

R8: Thank you for your comment. We have re-analyzed and simulated the peaks of the XPS data in **Fig. R6**, and the revised manuscript is presented in **Supplementary Fig. 6**.

Fig. R6. (a) C_{1s}. (b) N_{1s} XPS spectrum of TpDPP-Py COFs.

C9: The lattice parameters may not suitable for the symmetry of TpDPP-Py COFs.

R9: Thank you for your comment. We have re optimized and adjusted, and updated the lattice parameters suitable for TpDPP Py COFs: $a = 49.06 \text{ \AA}$, $b = 40.77 \text{ \AA}$ and $c = 3.88 \text{ \AA}$. $\alpha = 89.88^\circ$, $\beta = 89.84^\circ$ and $\gamma = 90.09^\circ$. The cell parameters of TpDPP-Py COFs are shown **Supplementary Table 1**.

C10: The authors are required to replace the Chinese text in Figure S7a Figure S9b with English.

R10: Thank you for your suggestion. We have changed the Chinese text into English.

C11: Please optimize the layout of Table S1 in the Supporting Information for improved readability.

R11: Thank you for your suggestion. We have replaced **Table S1** with **Table S3** and further optimized it.

C12: The author mentions that “it is the best performance among the reported materials” in page 1 line 27, however, in Table S1 of the Supporting Information, reference 10 has already achieved 99% conversion and 100% selectivity. The author needs to reconsider their statement.

R12: Thank you for your suggestion. For reference 10, TFB-33-DMTH exhibits 99% conversion and 100% selectivity in 20 h. However, the TpDPP-Py COF shows the photocatalytic conversion of 99% and selectivity of 98% in 20 min. The reaction rate of TpDPP-Py COF is significantly higher than that of TFB-33-DMTH. Anyway, to avoid misunderstandings, we have revised that the performance is better than most active material. The manuscript has been revised and marked in red color on page 9, line 19-20.

C13: In the line 77, the authors state that TpDPP-Py COFs exhibited the best catalytic performance compared with other reported materials. It might be overly definitive and exaggerate in nature.

R13: Thank you for your suggestion, we have deleted this sentence.

C14: In Figure 4a, the authors have indicated that the bandgap of TpDPP-Py COF is

1.38 eV. It is necessary to provide the calculation formula, as observed in Figure 4e and f, the calculated bandgap is 1.28 eV, with a 0.1 eV error between the calculated value and the actual value.

R14: Thank you for your suggestion, we have added the calculation formula. The revised reaction was marked in red color on page 7 line 8-9.

C15: The stability of the catalyst is also an important standard for evaluating performance. The authors should supplement the characterization results such as infrared and XRD after the catalyst has been cycled, to prove that the catalyst has not undergone significant changes.

R15: Thank you for your suggestion. The XRD pattern of the photocatalyst after the reaction has been provided, confirming that the crystal structure has not undergone changes as shown in **Fig. R7**. The revised manuscript has been highlighted in red on page 9, lines 24-26, and **Supplementary Fig. 22**.

Fig. R7. (a) XRD pattern of TpDPP Py COFs before photocatalytic reaction. (b) XRD pattern of TpDPP Py COFs after photocatalytic reaction. (c) FT-IR spectra of TpDPP Py COFs after photocatalytic reaction.

C16: The authors need to consider the clarity and aesthetics of the figures.

R16: Thank you for your suggestion, we have readjusted the clarity and aesthetics of the figures.

Responses to the reviewer 3:

In this manuscript, Fang et al. developed a new COFs with a wide absorption range, which significantly promotes the sunlight utilization efficiency for the artificial photocatalytic amine coupling. The thiophene functional groups were introduced into the donor-acceptor units for the synthesis of new TpDPP-Py COFs, exhibiting a wide absorption range from 200 nm to 900 nm (ultraviolet-visible-near infrared light) and two-photon fluorescence emission as well as Three-photon fluorescence emission, which is a major accomplishment for COFs molecular design and photocatalysis field. The photocatalytic performance with ~ 100% photocatalytic conversion rate is exciting. Overall, the results reported in this manuscript are quite interesting and suitable for publication in Nature Communications after minor revisions.

Answer: Thank you very much for your very positive comments. We have revised our paper according to your suggestions.

Specific comments:

C1: Donor and acceptor unit is important for the absorption of COFs. How to judge the ability of acceptor is not clear statement. For example, in figure 1c and 1d, strong or weak acceptor might be not appropriate. It is better to delete this. If you want to compare it, it is better to compare the LUMO levels of two different acceptors.

R1: Thank you for your suggestion. We have deleted this sentence and made modifications as shown in **Fig. R8**. The revised manuscript has been updated in **Fig. 1**.

Fig. R8. Design and synthesis of D-A COFs. **a, b** Synthetic route of TpDPP and DPP. **c, d** Molecular structure of TpDPP-Py COFs and DPP-Py COFs.

C2: Optoelectronic transmission dynamics data are missing in the article, the time-resolved photoluminescence spectra of TpDPP-Py COFs are suggested to supplement

them.

R2: Thank you for your suggestion. We have included the time-resolved photoluminescence spectrum of TpDPP-Py COFs with an excitation wavelength of 650 nm as shown in **Fig. R9**, confirming that TpDPP-Py COFs exhibit faster optoelectronic transport kinetics. The revised manuscript includes this data in **Supplementary Fig. 17 (b)**.

Fig. R9. Time-resolved photoluminescence spectra of TpDPP -Py COFs ($\lambda_{ex} = 650$ nm).

C3: The importance of thiophene functional groups was proposed in this study, but there is a lack of complete supporting data.

R3: Thank you for your feedback. The advantages of incorporating thiophene functional groups are clearly evident when comparing the performance of the two materials based on experimental data. Furthermore, we have included relevant literature, such as "*Angew. Chem. Int. Ed.* **62**, e202309624 (2023)," "*Phys. Chem. Chem. Phys.* **20**, 22997 (2018)," and "*J. Phys. Chem. C* **124**, 27403–27412 (2020)," to emphasize the significance of thiophene functional groups and strengthen our argument. The revised manuscript has been highlighted in red on page 3, lines 20-26.

C4: The author claims that the crystal structure of TpDPP-Py COFs is A-A stacking, as the comparison, the A-B stacking structure of simulated COFs should be provided and further sufficient explained.

R4: Thank you for your suggestion. We have simulated the A-B stacking structure of TpDPP-Py COFs and provided a detailed explanation in the supporting information as shown in **Fig. R10**. Subsequent to the modifications, **Supplementary Fig. 11** has been included in the supplementary information. The revised manuscript has been marked in red on page 6, lines 1-3.

Fig. R10. (a) Experimentally obtained PXRD powder pattern of TpDPP-Py COFs (purple), simulated pattern with AB stacking (black) and the bragg positions (green). (b) Simulated model TpDPP-Py COFs.

C5: The reason for the high photocatalytic efficiency of TpDPP Py COFs as short-lived materials should be further explained in detail.

R5: Thank you for your comment. We have further explained the short-lived materials in detail and cited references “*Nat. Synth.* **2**, 557–563 (2023), *J. Am. Chem. Soc.* **140**, 4623–4631 (2018)”. The revised manuscript was marked in red color on page 8 line 2-6.

C6: The mechanism of artificial photocatalytic amine coupling is also important, the author should provide a detailed explanation.

R6: Thank you for your suggestion. We have further explained the mechanism of artificial photocatalytic amine coupling and added a schematic diagram in **Fig. R11**. The revised manuscript includes a detailed explanation of the mechanism on page 11, lines 13-17, and page 12, lines 1-4, highlighted in red, and is presented in **Supplementary Fig. 25** in the supplementary information.

Fig. R11. Schematic diagram of the mechanism for photocatalytic selective conversion of BA to N-benzylidenebenzylamine.

C7: The authors should explain why “Artificial’ is added in the title of the manuscript.

R7: Natural photosynthesis is that the green plants (including algae) absorb light energy and synthesize energy rich organic compounds from carbon dioxide and water. Artificial Photosynthesis is that Artificial Photosynthesis is that Artificial synthesized photocatalysts for simulating natural photosynthesis, absorb light energy to produce high value-added products (including H₂, CO₂RR, NH₃, organic coupling et al.). The “Artificial” are widely used in the field of photocatalysis (*Nature Catalysis*, 2023, 6,464-475; *Nature Reviews Chemistry*, 2018, 2 160-173.; *Nature Communications*, 2023, 14, 5742.).

C8: It is recommended to correct some minor errors in spelling, grammar, format, etc., to make the article more professional and readable.

R8: Thank you for your suggestion. The English have been further polished and checked.

REVIEWERS' COMMENTS

Reviewer #1 (Remarks to the Author):

This manuscript can be accepted now. Nice work by authors.

Reviewer #2 (Remarks to the Author):

The paper has undergone significant improvements after the first revision and is now suitable for acceptance in Nature Communications without any further modifications.

Reviewer #3 (Remarks to the Author):

The authors have fully addressed my concerns. I can recommend this manuscript to Nature Communications now.